# Core Temperature Measurement—Principles of Correct Measurement, Problems, and Complications

**DOI:** 10.3390/ijerph182010606

**Published:** 2021-10-10

**Authors:** Hubert Hymczak, Aleksandra Gołąb, Konrad Mendrala, Dariusz Plicner, Tomasz Darocha, Paweł Podsiadło, Damian Hudziak, Radosław Gocoł, Sylweriusz Kosiński

**Affiliations:** 1Department of Anesthesiology and Intensive Care, John Paul II Hospital, 31-202 Krakow, Poland; hymczak@op.pl; 2Faculty of Medicine and Health Sciences, Andrzej Frycz Modrzewski Krakow University, 30-705 Krakow, Poland; 3Faculty of Medicine and Dentistry, Pomeranian Medical University, 70-204 Szczecin, Poland; olagoab13@gmail.com; 4Department of Anaesthesiology and Intensive Care, Medical University of Silesia, 40-055 Katowice, Poland; k.mendrala@gmail.com (K.M.); tomekdarocha@wp.pl (T.D.); 5Department of Cardiovascular Surgery and Transplantation, John Paul II Hospital, 31-202 Krakow, Poland; 6Institute of Medical Sciences, Jan Kochanowski University, 25-369 Kielce, Poland; p.podsiadlo.01@gmail.com; 7Department of Cardiac Surgery, Upper-Silesian Heart Center, 40-055 Katowice, Poland; damhud@gmail.com (D.H.); gocot@poczta.onet.pl (R.G.); 8Faculty of Health Sciences, Jagiellonian University Medical College, 31-008 Krakow, Poland; kosa@mp.pl

**Keywords:** thermoregulation, core temperature, measurement

## Abstract

Core temperature reflects the temperature of the internal organs. Proper temperature measurement is essential to diagnose and treat temperature impairment in patients. However, an accurate approach has yet to be established. Depending on the method used, the obtained values may vary and differ from the actual core temperature. There is an ongoing debate regarding the most appropriate anatomical site for core temperature measurement. Although the measurement of body core temperature through a pulmonary artery catheter is commonly cited as the gold standard, the esophageal temperature measurement appears to be a reasonable and functional alternative in the clinical setting. This article provides an integrative review of invasive and noninvasive body temperature measurements and their relations to core temperature.

## 1. Introduction

Thermoregulation is the element of homeostasis that maintains a stable internal temperature, regardless of the environmental conditions. Central thermoregulatory management is based on afferent thermal signals coming from the core and peripheral thermoreceptors, which are integrated by the central nervous system, especially the hypothalamus. Currently, the traditional model of an individual set point in the hypothalamus is being questioned. There is increasing evidence that body temperature is controlled by independent thermoeffector loops composed of specific afferent and efferent branches collectively producing a balance point of core temperature (T_c_) [1,2]. It is also believed that mean body temperature characterizes the overall thermal status, but its use is impractical for everyday use [3].

The core thermal compartment is very well perfused and relatively homogeneous. The T_c_ of the human body is maintained within a narrow range, close to 37 ± 0.5 °C at rest. Among healthy persons, the average daily temperature can differ by 0.5 °C. The menstrual cycle is also one of the main causes of temperature variations. In addition, daily variations can be as much as 0.25–0.5 °C, with the lowest body temperature occurring usually at about 4 a.m. and the highest about 6 p.m. [4]. Temperatures of the peripheral parts of the body are cooler than the core, depending on the environment and thermoregulatory vasoconstriction. Thus, the temperature that best describes a human thermal status is the T_c_. In the present review, we discuss the current insights into proper body temperature measurements as well as related challenges and complications.

## 2. Mechanisms of Thermoregulation

The T_c_ is a balance between heat generated by the body and heat lost to the environment. Humans are warm-blooded animals; therefore, our bodies are able to adjust their metabolic rate to maintain equal heat production and loss. This statement is important because proper body function depends on a stable T_c_. Body heat production is mainly the effect of a transformation of chemical energy in foods into heat by cellular oxidative metabolism [5,6].

Skin is the largest structure of the human body and is responsible for about 90% of heat loss. Heat loss rate is determined by the level of heat transferred from inner tissues to the skin and from the skin to the surrounding environment by four mechanisms: radiation, evaporation, convection, and conduction.

Radiation is the electromagnetic energy radiated by the body in the form of electromagnetic waves (mainly infrared rays) and is similar to the process of heat leaving a woodstove. Radiation is the most important mechanism of heat loss, accounting for approximately 65% of loss. This normal process usually occurs in air temperatures below 20 °C. Evaporation is an endothermic process that converts liquid into gas and is the second major source of heat loss, accounting for about 20%. Water vaporization needs energy and consumes heat, therefore accelerating heat loss. Evaporation is the basic mechanism of heat loss in high ambient temperatures. During intense exercise, the body loses 85% of its heat through sweating. Convection is heat loss by the physical movement of a fluid or gas from one location to another. This mechanism of heat loss is similar to sitting in front of a fan. The body loses 10% to 15% of its heat through convection. Conduction is the loss of molecular kinetic energy and occurs when heat is transferred between two objects that are in contact with one another, such as heat loss that occurs from sleeping on cold ground. The body loses about 2% of its heat through air conduction, although different mediums transfer heat by conduction at different rates. For instance, the conductive transfer of water is 100 times that of air, so heat can be lost from the body very quickly when it is immersed in cold water [5,6,7].

To maintain a normal T_c_, the hypothalamus works by a variety of autonomic mechanisms, such as changing basal metabolism, muscle tone, thyroid activity, or adrenal response. There is also a hypothalamic efferent component that includes the proper adjustment of sympathetic stimulation to subcutaneous vessels, thus inducing the vasoconstriction or vasodilation of peripheral vessels where heat loss is at its highest. Thermoregulatory functions can generally be divided into autonomic and behavioral reactions.

The primary autonomic thermoregulatory defenses are active precapillary vasodilation, arteriovenous shunt vasoconstriction, and shivering in cold exposure. Sweating (and consequent evaporation) is the only method of cooling down in a hot environment when the external temperature exceeds the temperature of the body. Sweating starts immediately after the onset of exercise, which is most likely due to various central mechanisms, including the baroreceptor reflex [8]. With heat acclimatization, sweat volume can be as high as 2–3 l/h, mainly due to lowering the temperature threshold for sweating and the redistribution of sweat secretion more peripherally [9].

In a cold environment, thermoregulatory vasoconstriction is mainly limited to arteriovenous shunts in the limbs. Although anatomically limited to toes and fingers, this mechanism can influence the blood flow to whole extremities and is efficient at cooling the body’s temperature when open and restricting metabolic heat when closed [3]. Shivering can quickly enhance the metabolic rate and continue to do so for up to 6 h [10]. Non-shivering thermogenesis, i.e., the activation of brown fat, can be used in infants and, in a small proportion, also in a certain adult population [11].

Other changes include an efferent behavioral component that is the most effective reaction to fluctuations in body temperature. Behavioral modifications initiated by excessive heat loss include adding more clothing, seeking shelter, curling up to decrease the proportion of exposed body surface area, exercising, or starting a fire to enhance heat gain via radiation. If the body is overheated, a common behavioral response would be to remove some outer clothing, minimize physical activity, or increase cooling by convection, such as standing in front of a fan [5,6].

To maintain the above mechanisms in hemostasis, the organism needs an appropriate cardiovascular function and intravascular volume, as the organism must be able to transfer the increasing internal temperature to its surface for release. Older persons are at increased risk for disturbances of thermoregulation as a result of reduced cardiac function and decreased intravascular volume. Behavioral reactions can be impaired in the elderly, especially for those with coexisting dementia, whereas, in newborns, numerous characteristics of their physiology contribute to the enhanced risk of heat loss. The large surface-area-to-body-mass ratio promotes cooling through conduction. Furthermore, newborns have a lesser amount of subcutaneous fat to provide insulation. In newborns, the blood flow is additionally modified, causing peripheral cyanosis. Finally, newborns lack the shivering mechanism and have to depend on non-shivering thermogenesis [12].

An abnormality in muscular function, reduced metabolism, or prolonged exposure to low ambient temperatures could result in hypothermia. At a body temperature below 33 °C, oxygen consumption decreases and enzyme activity reduces, whereas extended exposure to elevated temperatures, failure to disperse excessive body heat, or an increase in heat production (for example, enhanced metabolism or muscular work) may result in hyperthermia. During hyperthermia, the rate of reaction will diminish due to the denaturing of proteins, resulting in impaired cellular function [6,13].

## 3. Core Temperature Measurements

The precision of temperature monitoring depends on the measurement site and the measurement system. Four monitoring sites are considered to be T_c_ sites: the tympanic membrane, nasopharynx, esophagus, and pulmonary artery [3]. In clinical practice, these measurement locations are most preferable [3,14].

### 3.1. Body Surface

Due to poor peripheral circulation, the measurements do not accurately reflect the T_c_ but reflect only the temperature of the skin. Body surface measurement is performed using a thermistor that is applied to the surface of the skin or using an infrared thermometer. Numerous studies have demonstrated that the measurement of skin temperature using thermophototropic liquid crystals, typically used for forehead application devices, is imprecise and often shows a normal temperature despite an elevated T_c_ [6,13].

### 3.2. Oral

This is the most widespread technique for assessing body temperature. However, oral temperature is an unreliable method for the measurement of T_c_ because it is affected by environmental factors and bias caused by such factors as smoking or hot or cold beverages being consumed prior to the measurement. Lastly, mouth-to-mouth cross-infection or oral mucosa laceration may appear. Additionally, there are differences in the measurements recorded from the different sites in the mouth cavity, depending on the exact location of where the bulb of the thermometer is positioned. The preferred place for oral temperature measurement is the sublingual pocket [6,13,15].

### 3.3. Axilla

Measuring temperature in the axilla is less precise in comparison with the other noninvasive sites, and the temperature reading is usually much lower than the T_c_. Although it is safe, easily accessible, and a comfortable method of temperature measurement, there are numerous disadvantages. Axilla temperature measurement needs supervision in case the dislocation of the thermometer occurs, and this method is more time consuming than in other measurement sites. In addition, the measurement could be affected by ambient temperature, sweating, and evaporation. Therefore, this method of temperature measurement is not a suitable method to use in clinical settings [6,13].

### 3.4. Tympanic Membrane

This technique is quick and simple to use. Performed correctly, the reading is only slightly affected by environmental temperature. Moreover, it is safer than oral or rectal thermometers, especially for children. However, as the tympanic membrane receives blood from the branches of the internal carotid artery that supplies blood to the hypothalamus, this method of temperature measurement is problematic and an unreliable reflection of the T_c_. Being in direct contact with the environment, measurements at this location may be influenced by a lack of complete isolation from the outside temperature, obstruction or contamination of the auditory canal, or the presence of snow in the auditory canal. Moreover, in certain clinical conditions, such as cardiac arrest and reduced blood flow through the ICA, the tympanic membrane is not sufficiently perfused, and, therefore, temperature measurements can be inaccurate [13,16,17,18].

### 3.5. Rectum

Rectal temperature is in poor correlation with T_c_, and the readings can be significantly delayed [19]. Clinically, the rectal temperature is the most widely used, particularly in children. The rectal probe must be introduced to a depth of ≥15 cm so that the temperature sensors can be positioned near the large arteries of the pelvic region. Usually, temperature readings in the rectum are higher than those determined in other parts of the body. Moreover, in circulation instability, rectal temperature will be lower than the real T_c_. This kind of measurement is a reliable method only in conditions closely related to normothermia, but there is a considerable delay in rectal temperature results, particularly during rapid temperature changes. Moreover, rectal inflammation can influence temperature readings, and hard feces can impede the placement of the probe. Additionally, rectal temperature can be incorrectly higher if warmed peritoneal lavage is performed in patients with hypothermia. The insertion of the rectal probe can cause considerable pain and discomfort, especially for patients with perirectal infection. Conscious patients may experience a feeling of rectal fullness and the desire to defecate during insertion of the thermistor into the rectum. Moreover, rectal insertion is often terrifying and can be psychologically harmful for children. The risk of rectal perforation is another limitation of this method, especially for unconscious patients in whom distension and the thinning of the rectal wall develops due to the temporary paralysis of the autonomic nervous system [16].

### 3.6. Urinary Bladder

Kidneys receive about 25% of the cardiac output; therefore, if the urine flow rate is within a normal range, the urinary bladder temperature would closely match the T_c_. Since urine output is usually monitored in hospitalized patients, a simultaneous urinary bladder temperature measurement is very useful and is becoming increasingly common for use in intensive care unit patients. Temperature-sensing indwelling urinary catheters allow for the continuous drainage of urine and the constant measurement of body temperature. 

Due to the closeness of the urinary bladder to the rectum, the values of bladder temperature have characteristics very comparable to those of rectal temperature. Similar to the rectum temperature measurement, this technique is a reliable approach, but only in situations when body temperature is near to normothermia, whereas bladder temperature responds to T_c_ changes faster than rectal and skin temperature changes but slower than esophageal temperatures. A better correlation with T_c_ has been noted in the occurrence of high urinary volumes. Reduced urine production due to a decreased cardiac output or hypothermia causes this method of temperature measurement to be inaccurate. Similar to rectal temperature, bladder temperature can also be incorrectly higher if warmed peritoneal lavage is performed in patients with hypothermia. The problems and complications of this method of temperature measurement are the same as those that arise during bladder catheterization, of which catheter-associated urinary tract infection is the most common. Urethral injury, catheter obstruction, and urine leakage are other possible complication [5,16,20].

### 3.7. Nasopharynx

The nasopharynx is one of the most reliable locations for the measurement of T_c_. Additionally, the insertion of the probe into the nasopharynx is quite simply achieved and is a safe procedure. Therefore, the nasopharynx is a frequently used temperature monitoring location during surgical procedures. The optimal site for nasopharyngeal temperature measurement is near the internal carotid artery. The nearest part of the nasopharyngeal mucosa to the internal carotid artery is within the upper- or mid-nasopharynx. The nasopharyngeal probe should be inserted through the middle or inferior meatus in the nasal cavity. Any nasopharyngeal temperature probe insertion depth between 10 and 20 cm corresponds well to the T_c_ in adults [21,22,23]. Similar to tympanic membrane temperature measurement, this technique may give false low values in patients with unstable circulation and can lead to errors in the temperature measurements as a result of imprecise probe location or obstructed nasal canals.

### 3.8. Esophagus

The esophagus is the preferred location to determine T_c_ because of its location near the left atrium and left ventricle. Esophageal temperature strongly associates with pulmonary artery temperature and is a basic method of temperature measurement for intubated patients. The determination of esophageal temperature is also favored because of its quick reaction to changes in T_c_ [17,20,24].

Esophageal temperature measurement should reveal the temperature of the myocardium and, therefore, the probe must be perfectly placed at the level of the heart. The esophageal probe must be inserted in the lower third of the esophagus, and a lateral chest X-ray needs to be taken for an evaluation of the insertion length of the probe. An assessment of chest X-rays demonstrates that, at the level of the T8 and T9, the probe is under the tracheal bifurcation and close to the heart. Several non-radiographic methods of assessing the position of the esophageal probe have also been proposed. These methods are usually based on the height of the patient [13,16,17,19]. The following equations were developed to calculate the distance from the nasal flare to the point between T8 and T9, based (L) on the patient’s height: L (cm) = 0.228 × standing height − 0.194, or more simply, L (cm) = standing height/5 + 5 cm [25].

Although esophageal probe insertion is usually considered to be a safe method, there is the possibility of some complications. Major complications are rare and include esophageal bleeding, perforation, and arrhythmias. Temperature measurements taken with the use of a proximally positioned probe can be incorrectly elevated owing to ventilation with warmed gases. Furthermore, fluids passing through the nasogastric tubes can also change the temperature indication. Malposition of the esophageal probe can cause inaccurate T_c_ readings and cause bronchospasm and hypoxemia due to tracheal misplacement.

### 3.9. Pulmonary Artery

The use of the pulmonary artery catheter is limited to a specific group of patients, mainly critical care and cardiac surgery patients, to monitor hemodynamic parameters, such as cardiac output, pulmonary artery wedge pressure, and vascular resistance. Additionally, it is possible to measure blood temperature with a thermistor placed on the distal port in the pulmonary artery. T_c_ measurement with the pulmonary artery catheter is the most precise technique in clinical settings because the pulmonary artery carries blood directly from the body’s core; however, this technique is too invasive for routine use [24,26]. 

A study comparing pulmonary artery temperature measurement to other methods showed that infrared ear thermometers offered a comparatively close assessment of pulmonary artery T_c_, but with more changeability than oral or urinary bladder techniques, whereas axillary measurements were significantly lower than the pulmonary artery temperature and extremely variable [27]. The insertion of a Swan–Ganz catheter, which is used for pulmonary artery temperature measurement, can lead to several complications, including pneumothorax, cardiac arrhythmias, pulmonary infarction, and infection [28].

## 4. Conclusions

A reference technique for T_c_ measurement is the pulmonary artery temperature reading. In clinical settings, the temperature readings obtained from the esophagus are considered to be a gold standard, whereas tympanic measurement using the thermistor technique is a reliable option and is a suitable alternative if invasive methods of T_c_ measurement are impossible to achieve (Table A1) [26,29].

Ideally, the temperature measurement should accurately reflect the T_c_ in all age groups, be easy, non-invasive, harmless, and technique independent. Lastly, it should indicate the T_c_ as accurately as possible without being markedly influenced by the ambient temperature. Although we are in an era of continuing implementation of sophisticated medical technology, no standard techniques are currently being developed to determine body temperature. Twenty years ago, Moran and Mendal stated that, during this amazing era, we “forgot” to develop a better method for measuring body temperature [18,27]. This statement is still valid, and today we can also state that it is surprising that such an instrument has not yet been created. The key is selecting an appropriate monitoring site depending on clinical circumstances.

## Data Availability

Not applicable.

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
