# Peer review of "Core Temperature Measurement—Principles of Correct Measurement, Problems, and Complications"

_ijerph, 2021, doi:10.3390/ijerph182010606_

Round 1

Reviewer 1 Report

See attachment

Author Response

Dear Reviewer of International Journal of Environmental Research and Public Health,

Thank you very much for your patience and all the valuable comments and suggestions, which were very helpful to improve this manuscript. I tried my best to address all your concerns. Please find my answers below.

Sincerely yours,

Dariusz Plicner on behalf of the authors.

Major comments

  1. Introduction: You do not mention temperature changes due to menstrual cycle, which is a major variability factor of Tc.

Ad.1 The phrase: “The menstrual cycle is also one of the main causes of temperature variations” was added to the Introduction section, according to your suggestion (lines 43-44).

  1. L70-71: Convection does not only regard fluids but also gases, as you state with your “fan” example.

Ad.2 Thank you for your comment, the sentence has been amended (lines 69-70).

  1. L95-96: While correct, I find this formulation misleading because brown fat tissue is found in all infants whereas it is found only in < 10% of adults, with big sex differences.

Ad.3 The sentence has been modified, according to your suggestion (lines 97-98).

  1. L108: I don’t think one can stat that every elderly individual has impaired behavioral reactions. Consider reformulating this phrase.

Ad. 4 The sentence has been modified, according to your suggestion (lines 110-111).

  1. L116-117: I would argue that oxygen consumption already decreases at temperatures above 32°C. (see e.g., Grand et al., 2021, https://doi.org/10.1089/ther.2020.0013).

Ad. 5 Thank you for your suggestion, the quoted temperature threshold has been changed to 33 °C (line 119).

  1. L125-126: I would not say that these locations of Tc measurement are interchangeable (see e.g., Mase et al., 2021, https://doi.org/10.1186/s40560-021-00558-4).

Ad. 6 : Thank you for your comment, the sentence has been changed (lines 127-128).

  1. L155-156: Could you elaborate why receiving blood from the ICA is problematic? How is this not a problem when measuring Tc in the nasopharynx? (L206).

Ad. 7 Thank you for your comment,  we rewrote the paragraphs to make them more readable (lines 161-166 and 218-221).

  1. L244-L246: The indication for a pulmonary artery catheter is usually not Tc measurement. Consider rephrasing this phrase.

Ad. 8 Thank you for your comment, the phrase was  modified (lines 255-259).

  1. L262-263: You already state in L250-251 that oesophageal measurement is the gold standard.

Ad. 9 One of the sentences has been removed, according to your suggestion  (lines 285-286).

Minor comments:

  1. L93: the “are” should be an “is” as it refers to “mechanism”.

Ad 1 Thank you, it is done.

  1. L105: “have” should be “has” as it refers to “organism”.

Ad 2 Thank you, it is done.

Additionally, summary table was added.

Reviewer 2 Report

This mini-review prepared by the authors is not new, but I think it is an easy-to-understand review. However, the reviewer suggests the following two minor points.

1. You should make a clear figure or table in your mini-review to help the reader's understanding.

2. You should cite more recent original articles and reviews within the last five years.

Author Response

Dear Reviewer of International Journal of Environmental Research and Public Health,

Thank you very much for your patience and all the valuable comments and suggestions, which were very helpful to improve this manuscript. I tried my best to address all your concerns. Please find my answers below.

Sincerely yours,

Dariusz Plicner on behalf of the authors.

  1. You should make a clear figure or table in your mini-review to help the reader's understanding.

Ad 1 Thank you for your suggestion, we added the summary table.

  1. You should cite more recent original articles and reviews within the last five years.

Ad 2 Thank you for your suggestion, we added more recent references:

Nakamura K. Afferent pathways for autonomic and shivering thermoeffectors. Handb. Clin. Neurol. 2018, 156, 263-279, doi: 10.1016/B978-0-444-63912-7.00016-3.: 30454594.

Romanovsky AA. The thermoregulation system and how it works. Handb. Clin. Neurol. 2018, 156, 3-43, doi: 10.1016/B978-0-444-63912-7.00001-1.

Baker LB. Physiology of sweat gland function: The roles of sweating and sweat composition in human health. Temperature (Austin). 2019, 6, 211-259, doi:10.1080/23328940.2019.1632145.

Périard JD, Eijsvogels TMH, Daanen HAM. Exercise under heat stress: thermoregulation, hydration, performance implications, and mitigation strategies. Physiol. Rev. 2021, 101, 1873-1979, doi: 10.1152/physrev.00038.2020.

Haman F, Blondin DP. Shivering thermogenesis in humans: Origin, contribution and metabolic requirement. Temperature (Austin). 2017, 22, 217-22, doi: 10.1080/23328940.2017.1328999.

Chondronikola M, Bartelt A, Vidal-Puig A, Virtanen KA. Brown Adipose Tissue: From Heat Production in Rodents to Metabolic Health in Humans. Front. Endocrinol. (Lausanne). 2021, 12, 739065, doi:10.3389/fendo.2021.739065.

Urisarri A, González-García I, Estévez-Salguero Á, et al. BMP8 and activated brown adipose tissue in human newborns. Nat. Commun. 2021, 12, 5274, doi:10.1038/s41467-021-25456-z.

Masè M, Micarelli A, Falla M, Regli IB, Strapazzon G. Insight into the use of tympanic temperature during target temperature management in emergency and critical care: a scoping review. J. Intensive. Care. 2021, 9, 43, doi:10.1186/s40560-021-00558-4.